# The Role of Thoracic Ultrasonography and Airway Endoscopy in the Diagnosis of Equine Asthma and Exercise-Induced Pulmonary Hemorrhage

**DOI:** 10.3390/vetsci8110276

**Published:** 2021-11-15

**Authors:** Chiara Maria Lo Feudo, Luca Stucchi, Elena Alberti, Giovanni Stancari, Bianca Conturba, Enrica Zucca, Francesco Ferrucci

**Affiliations:** 1Equine Sports Medicine Laboratory “Franco Tradati”, Department of Veterinary Medicine, Università Degli Studi di Milano, 26900 Lodi, Italy; chiara.lofeudo@unimi.it (C.M.L.F.); elena.alberti@unimi.it (E.A.); enrica.zucca@unimi.it (E.Z.); 2Veterinary Teaching Hospital, Università Degli Studi di Milano, 26900 Lodi, Italy; luca.stucchi@unimi.it (L.S.); giovanni.stancari@unimi.it (G.S.); bianca.conturba@unimi.it (B.C.)

**Keywords:** horse, equine, thoracic ultrasonography, airways endoscopy, equine asthma, EIPH

## Abstract

Mild-moderate (MEA), severe (SEA) equine asthma and exercise-induced pulmonary hemorrhage (EIPH) are common respiratory disorders in horses. The present retrospective study aims to evaluate the role of ultrasonography and endoscopy in the diagnosis of these conditions. Three hundred and three horses were included and divided into SEA, MEA and MEA + EIPH groups, on the basis of history, clinical examination and bronchoalveolar lavage fluid (BALf) cytology; scores were assigned to lung ultrasonography, pharyngeal lymphoid hyperplasia (PLH), tracheal mucus (TM) and tracheal bifurcation edema (TB). These scores were compared between groups, and their associations with age, BALf cytology, tracheal wash microbiology and between endoscopic and ultrasonographic scores were statistically analyzed. Ultrasonographic scores were higher in the SEA and MEA + EIPH groups and associated with increased BALf neutrophils and hemosiderophages. The PLH score was higher in younger horses affected by MEA and EIPH and associated with increased eosinophils and hemosiderophages. TM and TB scores were greater in older horses affected by SEA, associated with increased neutrophils and inversely correlated with hemosiderophages. Moreover, TM grade was negatively correlated with mast cells. Thoracic ultrasonography and airway endoscopy can provide useful information about the inflammatory status of upper and lower airways in the horse.

## 1. Introduction

Equine asthma (EA) is a chronic and recurrent respiratory syndrome characterized by airflow obstruction, mucus hypersecretion and airway hyperreactivity [1]. Although some clinical presentations, such as chronic cough and poor performance, are common to asthmatic horses of all severities [2], EA can be classified as mild-moderate (MEA) or severe (SEA) [1]; while SEA typically affects adult horses over 7 years of age [3], MEA can affect horses of any age but is more commonly reported in young horses [4]. A diagnosis of EA can be reached by collection of history, clinical examination, airway endoscopy and cytology of the bronchoalveolar lavage fluid (BALf) and/or tracheal wash (TW) [2]. Some authors suggest that MEA could be associated with exercise-induced pulmonary hemorrhage (EIPH) [5], while other studies observed no significant correlation between these disorders [6,7,8,9]. Therefore, this represents a controversial topic. EIPH is a common disorder in racehorses and other sport horses subjected to strenuous exercise, defined as the detection of blood in the trachea at endoscopic examination after exercise [10] or the presence of hemosiderin in alveolar macrophages in BALf [11].

Some studies suggest that, in EIPH-affected horses, imaging may aid to diagnosis. Thoracic radiographs could show the presence of densities in the caudodorsal areas of the lungs; however, in many EIPH-affected horses only minimal radiographic abnormalities are observed and non-EIPH horses could have marked abnormalities too [12,13,14,15]. In one study, thoracic ultrasonography proved to be highly sensitive in detecting the presence of fluid accumulation in the caudodorsal lung fields in EIPH-affected horses; however, it has a low specificity, not allowing to distinguish between the presence of blood or inflammatory fluid [16]. In EA, imaging is not considered part of the minimum diagnostic protocol [2]. Radiographic abnormalities do not seem to be associated with BALf cytology or pulmonary function in MEA horses [17]. Endobronchial ultrasound seems to be promising in identifying airway smooth muscle remodeling in SEA-affected horses [18], but its use is still limited to research purposes. Only one preliminary study describes the use of thoracic ultrasonography for the diagnosis of EA, reporting that SEA-affected horses show more intense ultrasound changes compared to healthy horses and to MEA-affected horses; however, it only included a small number of patients [19]. 

Airway endoscopy is pivotal for the diagnosis of both EA and EIPH; in EA, the presence of mucus in the trachea is likely to predict airway inflammation [20] and, in particular, the detection of mucus grade ≥2/5 in racehorses or ≥3/5 in other sport or pleasure horses may be sufficient to diagnose EA [4]. The presence of pharyngeal lymphoid hyperplasia (PLH), visible at endoscopy, is associated with stabling and exposure to dust and other antigens, which are also associated to EA [21]. Moreover, in Thoroughbreds in training, an association between coughing, tracheal mucus accumulation and PLH has been observed [22]. However, no direct relationship between PLH and EA has been demonstrated yet [21]. Although it is reported that EA-affected horses may show thickening and blunting of the tracheal bifurcation [23], the results of a study about tracheal septum scoring revealed no differences between asthmatic and healthy horses [24]; therefore, the clinical relevance of this finding remains unclear. While the detection of blood in the airways after exercise is considered the gold standard for the diagnosis of EIPH [10], contrasting results have been obtained regarding the association between EIPH and tracheal mucus accumulation [5,6,8].

Since no consistent and solid results are reported about ultrasonographic and endoscopic findings in asthmatic and EIPH-affected horses, the present study aims to evaluate whether a complete ultrasonographic and endoscopic assessment could aid in the diagnosis of SEA, MEA and concomitant MEA and EIPH in horses.

## 2. Materials and Methods

### 2.1. Horses

For this retrospective study, the clinical records of the horses affected by SEA or MEA hospitalized at the Equine Unit of the Veterinary Teaching Hospital of the University of Milan (Italy) between 2001 and 2020 were reviewed. The patients that underwent a complete diagnostic evaluation, including clinical history, general physical examination, thoracic ultrasonography, airway endoscopy and BALF and/or TW cytology, were included in the study and divided into three groups: SEA group, MEA group and MEA + EIPH group. SEA horses were selected on the basis of history of exercise intolerance, dyspnea and coughing, normal rectal temperature and neutrophils count > 5% in BALF or >20% in TW cytology [25]. For MEA horses, inclusion criteria were history of poor performance and occasional coughing, eupnea at rest, normal rectal temperature and BALF cytology consisting of neutrophils > 5%, and/or mast cells > 2% and/or eosinophils > 1% [26]. Among the MEA horses, whenever hemosiderophages were observed in the BALF cytology and represented ≥10% of total alveolar macrophage count, the patients were considered as EIPH positive [16] and therefore included in the MEA + EIPH group.

### 2.2. Thoracic Ultrasonography

A thoracic ultrasonographic examination was performed on every horse before sedation and endoscopy. The ultrasound machine (Technos MPX, Esaote, Firenze, Italy; MyLab ClassC, Esaote, Firenze, Italy) was used with a 3.5 to 5.0 MHz convex transducer. In order to obtain high quality images, the image parameters were adjusted to every single patient. To increase acoustic coupling between the skin and the transducer, the skin was cleansed with alcohol and ultrasound gel was applied on the transducer. The ultrasound examination was performed bilaterally, from the 3rd intercostal space (ICS) to the 17th ICS, moving the transducer in a dorsoventral direction. The pleural surface was observed and eventual pathological alterations, such as comet-tail artifacts and wider focal lesions, were detected. To assign precise scores to the ultrasonographic findings, the lung area was divided into cranial, middle and caudal [19]: the cranial area included 3rd–7th ICSs, the middle area included 8th–12th ICSs and the caudal area included 13th–17th ICSs. A score was assigned to each area, on the left and on the right side, on the basis of the presence of comet-tail artifacts, using a specifically developed scoring system based on the one created by Siwinska and colleagues [19] (Table 1). The scores assigned to each area on one side were summed to those assigned to the same areas on the opposite side of the thorax. In this way, scores for cranial, middle and caudal areas were obtained. Then, the scores for each area were summed to calculate the total lung ultrasonographic score. Moreover, the number of focal lesions in each area was recorded. Similarly to the ultrasonographic scores, the numbers of focal lesions on both sides were summed to obtain the number for cranial, middle and caudal areas, and then the total number of focal lesions.

### 2.3. Airway Endoscopy, BAL and TW

To perform airway endoscopy and collect BALf or TW samples, a flexible videoendoscope (EC-530WL-P, Fujifilm, Tokyo, Japan) was used. The horses were previously sedated with detomidine hydrochloride (0.01 mg/kg IV; Domosedan; Vetoquinol, Italy) and restrained with a twitch. The endoscope was passed through the nasal passages and the upper and lower tracts of the respiratory system were examined. Endoscopic findings were graded using three scoring systems: a 0–4 score for pharyngeal lymphoid hyperplasia [27], a 0–5 score for tracheal mucus accumulation [28] and a 0--4 score for edema of the tracheal bifurcation [29]. After the visualization of upper and lower airways, BALf and TW samples were collected. Whenever the SEA horses presented severe dyspnea at the moment of the examination, only TW was performed. To perform the BAL, 60 mL of a 0.5% lidocaine hydrochloride solution was sprayed at the level of the tracheal bifurcation to inhibit coughing reflex. Then, the endoscope was passed into the bronchial tree until it was wedged firmly within a segmental bronchus based on the diameter of the endoscope, selected on the basis of the ultrasonographic findings. Here, a 300 mL pre-warmed sterile saline 0.9% was instilled and the fluid immediately aspirated [30]. The BALf sample was then stored in sterile ethylenediaminetetraacetic acid (EDTA) tubes and processed within 90 min. To perform the cytological examination, a few drops of pooled BALf were cytocentrifugated (ROTOFIX 32, Hettich Cyto-System, Germany) at 26× *g* for 5 min. The slides were air dried, stained with May-Grünwald Giemsa and Perl’s Prussian blue and observed under a light microscope at 400× and 1000× for 400-cell leukocyte differential count [31]. To perform the TW, a 60 mL pre-warmed sterile saline 0.9% was flushed into the intrathoracic portion of the tracheal lumen, re-aspirated and transferred into sterile plain tubes for microbiological evaluation [29]. To this aim, 10 microliters of pooled TW were cultured on blood agar plates 5% and incubated at 37 °C; after 48 h, bacterial species identification and UFC/mL count were performed [32]. In the SEA-affected patients that did not undergo the BAL, a cytological examination of the TW was performed.

### 2.4. Statistical Analysis

The data of the three groups (SEA, MEA, MEA + EIPH) were analyzed using descriptive statistics and evaluated for normality by means of the Shapiro–Wilk test. As data were not normally distributed in every group, non-parametric statistics was applied. The Kruskal–Wallis test, with Dunn’s multiple comparison test, was used to compare age, BALf leukocyte differential count, endoscopic scores, ultrasonographic scores (total lung area, cranial area, middle area and caudal area) and number of lung focal lesions (total and in every lung field) between groups. The differences between the ultrasonographic scores and the number of focal lesions on the left and right sides of the thorax were evaluated by means of the Mann–Whitney test. The association between age and BALf leukocyte differential count, endoscopic scores, ultrasonographic scores and number of lung focal lesions was evaluated using the Spearman correlation. The Spearman correlation was also used to evaluate the relationship between BALf leukocyte differential count and endoscopic scores, ultrasonographic scores and number of lung focal lesions. The Kruskal–Wallis test was used to evaluate the association of the endoscopic scores with the ultrasonographic scores and the number of lung focal lesions. The frequencies of positive results in the microbiological examination were compared between groups by means of Fisher’s exact test. The relationship between microbiological findings and age, BALf leukocyte differential cell count, endoscopic scores, ultrasonographic scores and number of lung focal lesions was evaluated using the Mann–Whitney test. Data are presented as median and interquartile ranges (IQR). Statistical significance was set at *p* < 0.05. Data were analyzed using a commercially available statistical software package (GraphPad Prism 9.1.0 for MacOS; GraphPad Software, San Diego, CA, USA).

## 3. Results

### 3.1. Horses

A total of 303 horses met the inclusion criteria and were divided into SEA group (62 horses), MEA group (118 horses) and MEA + EIPH group (123 horses). The study population consisted of 83 mares, 80 geldings and 140 stallions, aged from 2 to 28 years (median 5, IQR 3–9) and weighing from 272 to 680 kg (median 464 kg, IQR 434–507 kg). The most represented breeds were Standardbreds (180 horses), followed by Warmbloods (58 horses), Thoroughbreds (17 horses) and Quarter Horses (14 horses); the remaining 34 horses belonged to different mixed breeds. A significant difference was observed for age between the SEA group and the MEA group (SEA median 13 years, IQR 10–17 years; MEA median 4 years, IQR 3–8 years; *p* < 0.0001) and between the SEA group and the MEA + EIPH group (MEA + EIPH median 4 years, IQR 3–5 years; *p* < 0.0001), while no difference was detected between the MEA group and the MEA + EIPH group. 

### 3.2. Cytological Findings

Since 26 SEA horses were dyspnoic at the time of the examination, BAL and its cytology were performed in 277 horses (91.42%). Leukocyte differential cell counts of the BALfs of the SEA, MEA and MEA + EIPH horses are shown in Table 2.

When compared to the MEA and the MEA + EIPH groups, the SEA group had significantly greater percentages of neutrophils (*p* < 0.0001) and lower counts of macrophages (*p* < 0.0001), lymphocytes (*p* < 0.0001) and mast cells (SEA vs. MEA: *p* = 0.0035; SEA vs. MEA + EIPH: *p* = 0.0002). Moreover, the percentages of eosinophils were lower in the SEA group compared to the MEA + EIPH group (*p* = 0.0159). Significantly more hemosiderophages were observed in the MEA + EIPH group compared to the MEA and the SEA groups (*p* < 0.0001). No differences were observed between the MEA and the MEA + EIPH horses concerning the counts of macrophages, lymphocytes, neutrophils, eosinophils and mast cells, nor between the SEA and the MEA horses concerning the counts of eosinophils and hemosiderophages.

Age was associated to the percentages of all classes of BALf leukocytes (*p* < 0.0001). In particular, a positive correlation was observed with neutrophils (r = 0.5878), while an inverse correlation was detected with macrophages (r = −0.3394), lymphocytes (r = −0.4260), eosinophils (r = −0.2667), mast cells (r = −0.229) and hemosiderophages (r = −0.3969).

### 3.3. Microbiological Findings

A TW and its microbiological examination were performed in all horses. The bacterial culture was positive in 97 horses (32.01%) and negative in 206 horses (67.99%); in particular, the most frequently isolated bacteria were *Streptococcus* spp. (61 horses), *Pasteurella/Actinobacillus* spp. (16 horses) and *Klebsiella* spp. (7 horses). In the SEA group, positive results were observed in 20.97% of the horses, 36.44% in the MEA group and 33.33% in the MEA + EIPH group.

The frequencies of positive results in the MEA groups were significantly higher compared to the SEA group (*p* = 0.0418), while no significant differences were detected between the MEA + EIPH group and the SEA nor the MEA groups. 

The median age of horses having positive bacterial cultures was 4 (IQR 3–7) years old, while that of those having negative results was 5 (IQR 3–10) years old; horses with positive results were significantly younger than those having negative results (*p* = 0.0156).

There was no association between the results of microbiology and BALf cytology.

### 3.4. Ultrasonographic Findings

At ultrasonographic examination, the pleural surface was clearly identifiable, and scores could be assigned to every patient. Twenty-five horses (8.25%) showed a normal pleural surface, visible as a hyperechogenic line with underlying reverberation artifacts, while 278 horses (91.75%) presented some alterations. In particular, comet-tail artifacts were observed in 278 horses (91.75%), while focal lesions were present in 93 horses (30.69%). In total, 159 focal lesions were observed; of these, 86 were on the right side (54.09%), while 73 were on the left side (45.91%). However, no significant difference in the ultrasonographic score and number of focal lesions was detected between the two sides of the thorax. The ultrasonographic scores of the SEA, MEA and MEA + EIPH groups are shown in Figure 1. The median number of lung focal lesions was 0 in every group, in each area and in total. The IQR ranged from 0 to 1 in the cranial area and in total in the SEA and the MEA + EIPH groups; instead, it was 0–0 in the middle and caudal areas in every group, and in the cranial area and in total in the MEA group.

The MEA + EIPH group had significantly higher ultrasonographic scores of the cranial and caudal lung areas when compared to the MEA group (*p* < 0.0001), while no differences were detected between the SEA and the MEA nor the MEA + EIPH groups. In the middle area, no differences were detected between any groups. The MEA group showed a significantly lower total lung ultrasonographic score compared to the MEA + EIPH group (*p* < 0.0001) and to the SEA group (*p* = 0.0287); instead, no differences were observed between the SEA and the MEA + EIPH groups. The MEA + EIPH group showed significantly more focal lesions in the cranial area of the lung compared to the MEA group (*p* = 0.0019), while no difference was observed between the SEA and the MEA nor the MEA + EIPH groups; in the middle and the caudal areas, no differences were detected between any groups. The MEA group presented significantly less lung focal lesions compared to the MEA + EIPH group (*p* = 0.0019) and to the SEA group (*p* = 0.0475), while no differences were observed between the SEA and the MEA + EIPH groups. 

Age was not significantly correlated with the ultrasonographic scores nor with the number of lung focal lesions in any areas and in total.

### 3.5. Endoscopic Findings

Airways endoscopy was performed in all horses and endoscopic scores (pharyngeal lymphoid hyperplasia, tracheal mucus, tracheal bifurcation) were assigned to every patient. The endoscopic scores of the SEA, MEA and MEA + EIPH groups are shown in Figure 2.

The MEA and MEA + EIPH groups had significantly higher scores of pharyngeal lymphoid hyperplasia compared to the SEA group (*p* < 0.0001); moreover, the MEA + EIPH group had slightly higher PLH scores compared to the MEA group (*p* = 0.0451). The SEA group had significantly higher tracheal mucus scores compared to the MEA and MEA + EIPH groups (*p* < 0.0001); in contrast, no difference was observed between the MEA and the MEA + EIPH groups. The scores of edema of the tracheal bifurcation were significantly higher in the SEA group compared to the MEA (*p* = 0.0004) and the MEA + EIPH (*p* < 0.0001) groups; instead, no difference was observed between the MEA and the MEA + EIPH groups.

Age was positively correlated with tracheal mucus score (*p* < 0.0001, r = 0.37) and tracheal bifurcation score (*p* = 0.0004, r = 0.20) and inversely correlated with pharyngeal lymphoid hyperplasia score (*p* < 0.0001, r = −0.66).

### 3.6. BALf Cytology and TW Microbiology vs. Ultrasonographic Findings

The total lung ultrasonographic scores were positively correlated with the percentages of BALf macrophages (*p* = 0.0110, r = 0.1525), neutrophils (*p* = 0.0009, r = 0.20) and hemosiderophages (*p* < 0.0001, r = 0.25) and negatively correlated with the percentages of lymphocytes (*p* = 0.0001, r = −0.23). The ultrasonographic scores of the cranial area were positively correlated with the percentages of BALf macrophages (*p* = 0.0008, r = 0.20), neutrophils (*p* = 0.0014, r = 0.19) and hemosiderophages (*p* < 0.0001, r = 0.26) and negatively correlated with the percentages of lymphocytes (*p* < 0.0001, r = −0.25). The ultrasonographic scores of the middle area were positively correlated with the percentages of BALf neutrophils (*p* = 0.0095, r = 0.16) and negatively correlated with the percentages of lymphocytes (*p* = 0.0021, r = −0.18). The ultrasonographic scores of the caudal area were positively correlated with the percentages of BALf macrophages (*p* = 0.0162, r = 0.14) and hemosiderophages (*p* < 0.0001, r = 0.27). 

The total number of lung focal lesions was positively correlated with the percentages of BALf macrophages (*p* = 0.0275, r = 0.13), neutrophils (*p* = 0.0104, r = 0.15) and hemosiderophages (*p* = 0.0251, r = 0.13) and negatively correlated with the percentages of lymphocytes (*p* = 0.0032, r = −0.18). The number of lung focal lesions in the cranial area was positively correlated with the percentages of BALf macrophages (*p* = 0.0170, r = 0.14), neutrophils (*p* = 0.0307, r = 0.13) and hemosiderophages (*p* = 0.0128, r = 0.15) and negatively correlated with the percentages of lymphocytes (*p* = 0.0103, r = −0.15). No association was observed between the number of lung focal lesions in the middle and caudal areas and the BALF cytology.

Microbiological examination results were not correlated with the ultrasonographic scores nor with the number of lung focal lesions in any areas and in total.

### 3.7. BALf Cytology and TW Microbiology vs. Endoscopic Findings

Pharyngeal lymphoid hyperplasia scores were positively correlated with the percentages of BALf macrophages (*p* = 0.0019, r = 0.19), eosinophils (*p* = 0.0005, r = 0.21) and hemosiderophages (*p* < 0.0001, r = 0.29) and negatively correlated with the percentages of neutrophils (*p* < 0.0001, r = −0.25). 

Tracheal mucus scores were positively correlated with the percentages of BALf neutrophils (*p* < 0.0001, r = 0.36) and negatively correlated with the percentages of lymphocytes (*p* < 0.0001, r = −0.33), mast cells (*p* = 0.0486, r = −0.12) and hemosiderophages (*p* < 0.0001, r = −0.28).

Tracheal bifurcation scores were positively correlated with the percentages of BALf neutrophils (*p* = 0.0231, r = 0.14) and negatively correlated with the percentages of lymphocytes (*p* = 0.0255, r = −0.13) and hemosiderophages (*p* = 0.0002, r = −0.22).

Microbiological examination results were not correlated with any endoscopic scores.

### 3.8. Ultrasonographic vs. Endoscopic Findings

Pharyngeal lymphoid hyperplasia scores were not associated with the ultrasonographic scores nor with the number of lung focal lesions in any areas and in total.

The ultrasonographic scores of the caudal area were significantly higher in horses having lower tracheal mucus and bifurcation scores (tracheal mucus: *p* = 0.0134; tracheal bifurcation: *p* = 0.0218), while no association was observed with the ultrasonographic scores of the cranial area, the middle area and the total lung ultrasonographic scores. A greater total number of lung focal lesions was observed in horses showing higher tracheal mucus scores (*p* = 0.0498); in contrast, tracheal bifurcation scores were not associated with the number of lung focal lesions in any areas and in total.

## 4. Discussion

The current study presents and compares the findings of thoracic ultrasonography and airway endoscopy in 303 horses affected by SEA, MEA or concomitant MEA and EIPH. The SEA horses were significantly older compared to the MEA and the MEA + EIPH horses. In fact, SEA usually affects adult horses older than 7 years [3], MEA is more commonly reported in young horses [4] and EIPH typically affects racehorses in career [10].

In our study, BALf neutrophils percentages were greater in the SEA group, while eosinophils and mast cells counts were higher in the MEA and the MEA + EIPH groups. In fact, SEA is typically neutrophilic [2], while different cytological phenotypes of MEA are reported (eosinophilic, mastocytic, neutrophilic) [4]. Moreover, younger horses had greater percentages of eosinophils and mast cells, while neutrophils increased with age. This finding is in accordance with what was previously reported. In fact, the different MEA phenotypes have been associated with age [4]; in particular, the eosinophilic and mastocytic forms are more commonly observed in young horses (<5 years old) [4,33], while the neutrophilic form is particularly encountered in adult horses (>7 years old) [4]. Hemosiderophages were more numerous in younger horses, which can be easily explained by the fact that racehorses are the most frequently EIPH-affected patients [10] and usually have a short career [34].

Bacterial culture was positive in about one third of the examined horses and the most frequently isolated bacteria were *Streptococcus* spp., *Pasteurella/Actinobaciluus* spp. and *Klebsiella* spp.; β-haemolytic *Streptococci* and, in particular, *Streptococcus equi* subs. *zooepidemicus* are the most common opportunistic pathogens of equine airways, followed by *Pasteurella* spp., *Actinobacillus equuli*, *Klebsiella pneumoniae* and *Bordetella bronchiseptica*, which are frequently encountered in adult horses [29,35]. The role of bacterial infection or colonization in the pathogenesis of MEA has been widely discussed and investigated; however, variable results have been obtained by different studies. Multiple authors have observed an association between isolation of bacteria from the TW and MEA [36,37,38,39], presence of coughing [40] and tracheal mucus grade [37,41]. In particular, although *Streptococcus equi zooepidemicus* is the most strongly and consistently MEA-associated infectious agent [6,36,37,38,39,40,41], evidence of association with MEA has also been observed for *Actinobacillus/Pasteurella* spp. [37,38,39,40,41], *Streptococcus pneumoniae* [38,39,40] and *Bordetella bronchiseptica* [40]. However, different studies reported that the association between bacteria and MEA was significant only when total bacterial counts were greater than 10^3^ [39,40] or 10^4^ cfu/mL [36]. In fact, low bacterial counts could result from contamination of the samples, which is more likely in MEA horses that could cough during endoscopy and have more tracheal mucus, making the TW procedure longer and more complicated [40]. Even if an association between bacteria and MEA seems to exist, the causative role of infectious agents remains unclear. In fact, the prolonged colonization of the airways by these bacteria could be a consequence, rather than the cause, of the increased mucus accumulation in the trachea [38,41]. Moreover, the previously mentioned studies were mainly performed on young racehorses, while a recent investigation only including adult horses (over 8 years of age) observed no association between bacterial overgrowth and airway inflammation [42]. In our study, the MEA horses had more positive results to bacterial culture compared to the SEA horses. An explanation for this could be that MEA horses are younger than SEA horses and the isolation of bacteria was more frequent in young horses; this finding is in agreement with previous studies, reporting a decrease in the incidence of infections with age due to the development of immunity [37,38,39,43,44]. Even though we could have expected that horses with positive bacterial culture could show a higher BALf neutrophilia, compared to horses with negative microbiological results, no association was observed between the results of microbiology and cytology in our study; however, in the previous studies associating positive cultures to neutrophilia, cytology was performed on the TW [6], while in the current study it was performed on the BALf. This finding could suggest that BALf cytology, in the absence of pneumonia, is not influenced by bacterial colonization.

Ultrasonographic examination revealed that more than 91% of the horses included in this study presented comet-tail artifacts, similarly to what was observed in a previous study [19]; therefore, it is reasonable to consider that the detection of these alterations at ultrasonography could represent a highly sensitive tool to identify airway inflammation. However, the comet-tail artifacts lack specificity as they usually suggest the presence of a small amount of fluid surrounded by aerated lung [45], without allowing to distinguish the nature of the fluid (inflammatory fluid, edema, blood, etc.) [16,19]. Moreover, single comet-tail artifacts (less than 5) could also be observed in healthy patients, particularly in the lung periphery and the most cranioventral areas of the lung [45], as a result of previous asymptomatic infections or exposure to non-infectious inflammatory factors [19]. In our study, the MEA horses had less comet-tail artifacts and focal lung lesions compared to the SEA horses and those concomitantly affected by MEA and EIPH; therefore, the ultrasonographic appearance of the lung may provide fast and easy information on the severity of the inflammatory status of the lung. The current study did not include healthy horses, so it did not allow us to compare thoracic ultrasonography between healthy and asthmatic horses; however, a preliminary study on ultrasound changes in asthmatic horses observed a significant difference between SEA and healthy horses, while no differences were detected between MEA and healthy horses. Moreover, similarly to our study, it reported a significant difference between ultrasonographic findings in SEA and MEA horses [19]. Regarding the presence of EIPH, a previous investigation reported that ultrasonography had a high sensitivity but a low specificity; in particular, in EIPH horses the comet-tail artifacts were mainly located in the dorsocaudal lung portions [16], which are the areas most commonly affected by EIPH [46,47,48,49]. As expected, also in the present study, horses affected by MEA and EIPH showed more comet-tail artifacts in the caudal lung fields compared to the horses only affected by MEA. However, also in the cranial portions, the number of comet-tail artifacts and focal lesions was greater in the MEA + EIPH group compared to the MEA group. This finding could be explained by the fact that blood, even if it originates from the caudal areas of the lung, may accumulate for gravity in the cranioventral regions. 

In our study, although no significant difference was observed between the left and right side of the thorax for ultrasonographic score or number of focal lesions, it must be noticed that 54.01% of the focal lesions were observed on the right side. This finding is in agreement with the results of a previous study, where more advanced lesions were identified in the right lung fields [19]. In fact, it is reported that lung lesions are more common in the right side of the thorax [50,51]. This could be due to the fact that the right main bronchus is slightly wider and has a smaller bifurcation angle compared to the left bronchus, making the access of allergens and pathogens easier [19]. 

In this study, we found no association between age and ultrasonographic appearance. To date, the hypothetical relationship between thoracic ultrasound changes and ageing has not been studied in the horse. In human medicine, aging is related to decreased lung elasticity and compliance due to loss of elastin and collagen in the lung tissue [52,53]. In contrast, in the horse, alveolar walls do not seem to show any decrease in maximal extensibility in older patients [54]. 

The total ultrasonographic score proved, in our study, to be positively correlated with BALf neutrophils counts and percentages of hemosiderophages. In particular, neutrophils were correlated with the ultrasonographic scores in the cranial and middle lung fields, while hemosiderophages to the ultrasonographic scores in the cranial and caudal portions. Moreover, the total number of focal lesions was positively correlated with neutrophils and hemosiderophages counts, mostly in the cranial portions of the lung. These results are in accordance with the finding of more comet-tail artifacts and focal lesions in the SEA horses, in which neutrophilia is observed, and in the MEA + EIPH horses, which present higher counts of hemosiderophages compared to the MEA horses. Bacterial culture results were not associated with the ultrasonographic appearance of the lung. This could be due to the fact that, in our cases, a lung infection was not present, but the isolation of bacteria could only reflect a prolonged bacterial colonization of the airways.

The endoscopic scores for pharyngeal lymphoid hyperplasia were much lower in the SEA group compared to the MEA and the MEA + EIPH groups, while MEA + EIPH horses had greater scores compared to MEA horses. It has commonly been speculated that PLH may be associated with MEA [27,44,55]; however, no direct association has been proved and they just may coexist as a consequence to stabling and antigen exposure [56]. It has been suggested that severe PLH and other upper airway disorders could contribute to the pathogenesis of EIPH by compromising upper airway airflow and increasing the negative pressure in the lower airway during inspiration [57]; however, no association between PLH and EIPH has been demonstrated to date and a study reported that the severity of PLH did not have any effect on the incidence of EIPH [27]. In our study, PLH was also negatively correlated with age. In fact, it is thought that PLH may regress as the horse matures and develops immunity, and an inverse relationship between age and the prevalence of PLH has been reported by numerous studies [44,55,58,59,60,61]. 

The pathogenesis of PLH is unclear; however, it has been suggested that it may result as an immunological reaction to viral infections or secondary bacterial invasions or environmental antigens [55,56]; in our study, no association between bacterial culture and PLH score has been observed. Some authors reported that pharyngeal microflora varies based on the inflammatory status and, in particular, the number of bacteria per gram in pharyngeal secretions of horses affected by grade 3--4 LPH was about 100-fold higher compared to normal horses [62]. The divergence between these findings and our results could be explained by the fact that, in our study, microbiological evaluation was performed on the TW and a degree of independence between upper and lower airway environments has been demonstrated by previous investigations [59]. 

In our study, PLH was correlated with increased percentages of BALf eosinophils and hemosiderophages, while an inverse correlation was observed with the neutrophils counts. A negative correlation between neutrophils and PLH scores has also been observed by Koblinger and colleagues [59], while no association between PLH and cytology of the BALf and the TW was detected in other studies [56,60]. Our results can be easily explained. In fact, LPH decreases with age and is more common in horses affected by MEA and concomitant MEA and EIPH, eosinophils and hemosiderophages are more numerous in younger horses and horses affected by MEA and EIPH, while neutrophils increase with age and are abundant in SEA-affected horses. Moreover, PLH may represent a reaction to environmental antigens [55,56], and this would fit with BALf eosinophilia. 

The endoscopic scores for tracheal mucus and tracheal bifurcation were significantly higher in the SEA group, compared to the MEA and the MEA + EIPH groups, and increased with age. Tracheal mucus grade and tracheal bifurcation score have been reported to be related to each other [59]. The association between tracheal mucus score and age has been investigated by different authors with various results: in a study including only adult Warmbloods, no association has been observed [61]; in a study about young Thoroughbreds, mucus grade decreased with age [60]; and in another study including horses of various age and breeds, mucus score increased with age [59]. The variability of these results seems to be attributable to the different horse populations enrolled in the studies. Our findings, based on a diverse population, agree with those of the latter study which included equally various patients. Contrasting results have also been obtained regarding the association of tracheal bifurcation score and age. In fact, some authors report that tracheal bifurcation grade increases in older horses [24,28], while in another study no association with age has been detected; however, the same study reported greater tracheal bifurcation scores in the SEA-affected horses compared to the MEA-affected horses [59]. 

In our study, tracheal mucus and bifurcation scores were both positively correlated with BALf neutrophils counts and negatively correlated with hemosiderophages percentages. Moreover, tracheal mucus grade was also negatively correlated with mast cells counts. Mucus grade has been associated with increased neutrophils by different studies [28,59,60] and to decreased mast cells [59]; however, other authors report no correlation between tracheal mucus and BALf cytology [56,63,64]. It has been hypothesized that neutrophils may play an important role in airway mucus accumulation by inducing an increase in the production of mucus and a decrease in the clearance, while the role of mast cells in airway inflammation remains unclear [59]. The inverse correlation between tracheal mucus and hemosiderophages is in accordance with the findings of a previous study in which tracheal mucus was associated with a decreased likelihood of EIPH [8]; however, other authors reported no association between EIPH and mucus [4,10]. Only one study investigated the association between tracheal bifurcation and BALf cytology, reporting a negative correlation with mast cells [59]. 

In our study, no correlation between bacterial culture and tracheal mucus and bifurcation has been observed; however, it has been reported that the isolation of specific bacterial species, such as *Streptococcus equi zooepidemicus* and *Pasteurellaceae*, may be associated with increased mucus grade in racehorses [40,41,43]. It is not clear whether bacterial infections could contribute to mucus accumulation or bacterial colonization may be a consequence of impaired mucus clearance [4]. The lack of relationship between microbiology results and tracheal mucus in our study could be due to the fact that the horses having greater mucus grades were the SEA horses, in which less bacteria were isolated compared to the other groups.

In our study, some associations between ultrasonographic and endoscopic findings were observed. In particular, tracheal mucus grade was correlated with an increased number of lung focal lesions visible at ultrasound examination. Both these features have shown to be typical of a severe inflammation of the airways and have commonly been found in the SEA horses. Instead, an inverse correlation was detected between tracheal mucus and bifurcation grades and the number of comet-tail artifacts in the caudal lung fields. In fact, lesions in the caudal portions are more common in horses affected by EIPH, which was negatively associated with tracheal mucus and bifurcation scores.

## 5. Conclusions

In conclusion, ultrasonographic examination represents a useful and non-invasive diagnostic tool for horses affected by equine asthma and exercise-induced pulmonary hemorrhage, providing fast and easy information on the inflammatory status of the lung. Horses affected by SEA or concomitant MEA and EIPH have a significantly higher ultrasonographic score compared to horses affected by MEA only. Moreover, ultrasonographic alterations proved to be associated with increased BALf neutrophils and hemosiderophages counts and to increased tracheal mucus accumulation. However, the diagnostic limit of thoracic ultrasonography is represented by its low specificity, not allowing to distinguish the nature of the lesions. Therefore, airway endoscopic and BALf collection remain essential in the diagnostic protocol of these respiratory conditions. Thoracic ultrasonography should be considered when selecting the bronchus from which to collect the BALf as it provides further information on the morphology of the lung, the localization and the extent of the sites of inflammation. Moreover, in young, moderately asthmatic racehorses, lung ultrasonography could represent a reliable indicator to distinguish between EIPH-affected and non-affected patients.

## Figures and Tables

**Figure 1 vetsci-08-00276-f001:**
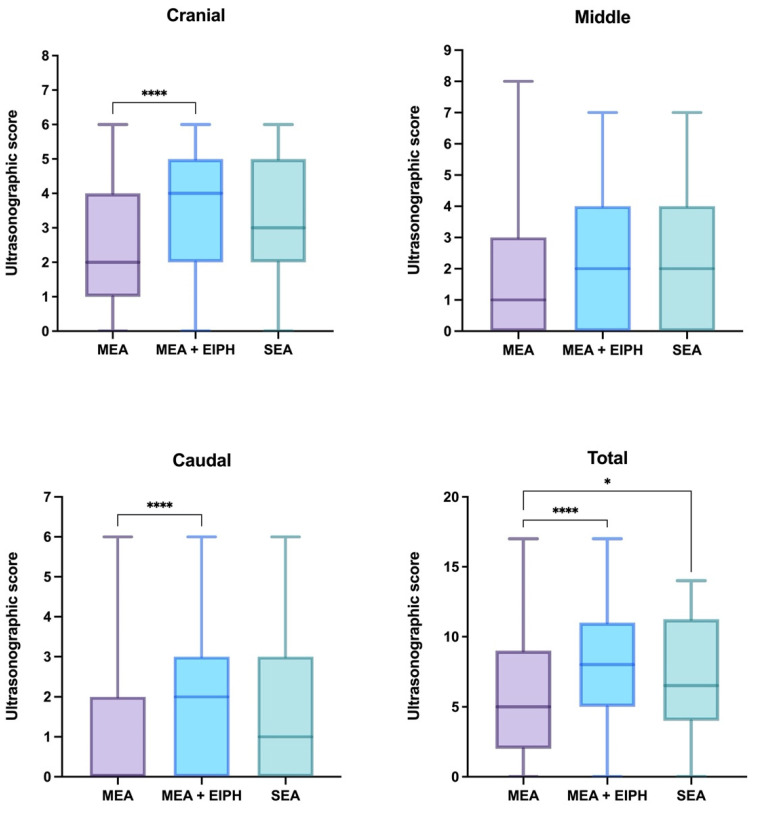
Box plot showing ultrasonographic scores of the cranial, middle and caudal lung fields, and the total ultrasonographic scores in the SEA, MEA and MEA + EIPH groups. The statistical significance is shown as * (*p* < 0.05) and **** (*p* < 0.0001).

**Figure 2 vetsci-08-00276-f002:**
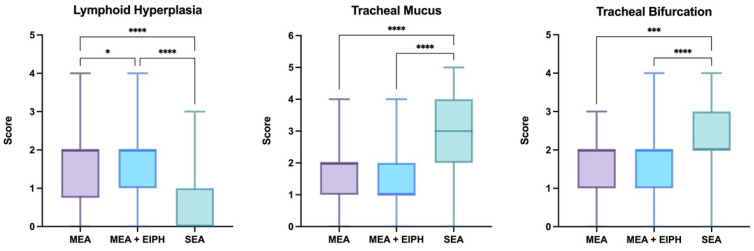
Box plot showing endoscopic scores of pharyngeal lymphoid hyperplasia, tracheal mucus and tracheal bifurcation in the SEA, MEA and MEA + EIPH groups. The statistical significance is shown as * (*p* < 0.05), *** (*p* < 0.001), **** (*p* < 0.0001).

**Table 1 vetsci-08-00276-t001:** Thoracic ultrasonography scoring system. Adapted from Siwinska et al., 2019 [19].

Score	Ultrasonographic Findings
0	Normal pleural surface, no comet-tail artifacts
1	Single comet-tail artifacts in 1 intercostal space
2	Numerous comet-tail artifacts in 1–2 intercostal spaces or single comet-tail artifacts in 2–3 intercostal spaces
3	Numerous comet-tail artifacts in 3–4 intercostal spaces
4	Numerous comet-tail artifacts in 5 intercostal spaces

**Table 2 vetsci-08-00276-t002:** Leukocyte differential counts, expressed as mean ± standard deviation, of the bronchoalveolar lavage fluids of the SEA, MEA and MEA + EIPH horses. Note: the percentages of the hemosiderophages are calculated out of the total of macrophages.

	SEA	MEA	MEA + EIPH
Macrophages	32.95% ± 13.18%	45.33% ± 10.17%	45.67% ± 8.82%
Lymphocytes	18.7% ± 10.68%	33.38% ± 12.78%	33.14% ± 10.84%
Neutrophils	44% ± 19.16%	14.44% ± 10.1%	13.4% ± 7.98%
Eosinophils	1.24% ± 2.24%	2.23% ± 3.63%	2.95% ± 4.32%
Mast cells	3.11% ± 1.85%	4.62% ± 2.47%	4.84% ± 2.39%
Hemosiderophages	0.62% ± 1.8%	2.49% ± 3.2%	24.26% ± 13.59%

## Data Availability

The data presented in this study are available on request from the corresponding author.

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
