# Peer review of "The Role of Thoracic Ultrasonography and Airway Endoscopy in the Diagnosis of Equine Asthma and Exercise-Induced Pulmonary Hemorrhage"

_vetsci, 2021, doi:10.3390/vetsci8110276_

Round 1

Reviewer 1 Report

Thank you very much for the manuscript on this interesting topic. It describes a study about the significance of thoracic ultrasonography and airway endoscopy in the diagnosis of equine asthma and exercise-induced pulmonary hemorrhage. The study covered a good cross section of the horse population. Unfortunately, a control group of horses without SEA, MEA and MEA + EIPH was missing. However, this issue is discussed in line 403 ff.  The scores of different parameters were raised and statistically analyzed. In conclusion, thoracic ultrasonography was described as complementary diagnostic method.

From my view there are only a few points to revise and reconsider:

  1. Line 36 spelling not correct “typically”
  2. Line 39 ff. very long sentence. Please rewrite it to improve the readability.
  3. Line 144 please convert rpm into g
  4. Line 239 ff. Please rewrite this sentence to improve the readability. For instance, use more shorter sentences.
  5. Line 344 spelling not correct “typically”
  6. Line 347 please correct the sentence “what was previously reported”
  7. Line 391 please correct the sentence “similarly to what was observed”
  8. Line 460 ff. please change the sentence to grammatically correct from “no association was been observed”

Author Response

Dear Reviewer,

Thank you very much for the appreciation and valuable advice. We have made some modifications to the text as suggested by you:

  • Line 36: spelling corrected
  • Line 39: we have modified this sentence in order to make it more readable
  • Line 144: rpm converted into g
  • Line 239: we have changed the punctuation to make it more readble
  • Line 344: spelling corrected
  • Line 347: sentence corrected
  • Line 291: sentence corrected
  • Line 460: grammar corrected.

We hope that the changes we made satisfy you.

Thank you again,

Dr. Chiara Lo Feudo

Reviewer 2 Report

I appreciate all the work that went into this study. Although it is a controversial topic, the authors have tried to make the subject objective using different scores. Perhaps, can be enrichment include in the discussion the possibilities of the pulmonary fibrosis associated with the ultrasonographic findings. In the ultrasonographic findings score only it has been included the comet tails, but there are not descriptions about small areas of consolidations that sometimes are visibles in this patients. The article is interesting and this topic could be developed in future experimental studies.

Author Response

Dear Reviewer,

Thank you for your appreciation and valuable advice. Although pulmonary fibrosis can be a feature of severely asthmatic horses, to our knowledge ultrasonographic alterations compatible with this have not been reported. Moreover, we have not included the areas of lung consolidation in the study because they are more frequently observed in horses affected by pneumonia, which were excluded. The only lung alterations we observed in our experience were comet-tail artifacts and wider focal lesions.

We hope these explanations satisfactorily answered your doubts.

Dr. Chiara Lo Feudo

Reviewer 3 Report

I was delighted with the article I received for review. Firstly, it deals with the problem of the clinical diagnosis of EA and EIPH with the use of commonly used methods and explains the doubts found during such studies. Secondly, it is based on a large group of patients, which makes the obtained results credible.
Critical analysis of the methods used in this retrospective experiment did not reveal any irregularities, in my opinion, therefore I recommend this article for publication in the presented form.

Author Response

Dear Reviewer, 

Thank you very much for the appreciation of our work. It is very rewarding.

Dr. Chiara Lo Feudo